# Wireless Epidermal Electromyogram Sensing System

**Sungjun Lee [1,2], Jiyong Yoon [3], Daewoong Lee [1,4], Duhwan Seong [3], Sangkyu Lee [3],
Minsu Jang [1], Junho Choi [1], Ki Jun Yu [2,*], Jinseok Kim [1,*], Sangyoup Lee [1,4,*] and
Donghee Son [3,*]**

[1] Center for Bionics of Biomedical Research Institute, Korea Institute of Science and Technology, Seoul 02792, Korea; leesj9883@kist.re.kr (S.L.); h15522@kist.re.kr (D.L.); minsujang@kist.re.kr (M.J.); junhochoi@kist.re.kr (J.C.)

[2] School of Electrical and Electronic Engineering, Yonsei University, Seoul 03722, Korea

[3] Department of Electrical and Computer Engineering, Sungkyunkwan University, Suwon 16419, Korea; jiyong428@g.skku.edu (J.Y.); dodoworld1993@gmail.com (D.S.); lee39@g.skku.edu (S.L.)

[4] Division of Bio-Medical Science & Technology, KIST School, Korea University of Science and Technology, 5, Hwarangro 14-gil, Seongbuk-gu, Seoul 02792, Korea

[*] Correspondence: kijunyu@yonsei.ac.kr (K.J.Y.); jinseok@kist.re.kr (J.K.); sangyoup@kist.re.kr (S.L.); daniel3600@g.skku.edu (D.S.); Tel.: +82-31-290-7696 (D.S.)

**Abstract:** Massive efforts to build walking aid platforms for the disabled have been made in line with the needs of the aging society. One of the core technologies that make up these platforms is a realization of the skin-like electronic patch, which is capable of sensing electromyogram (EMG) and delivering feedback information to the soft, lightweight, and wearable exosuits, while maintaining high signal-to-noise ratio reliably in the long term. The main limitations of the conventional EMG sensing platforms include the need to apply foam tape or conductive gel on the surface of the device for adhesion and signal acquisition, and also the bulky size and weight of conventional measuring instruments for EMG, limiting practical use in daily life. Herein, we developed an epidermal EMG electrode integrated with a wireless measuring system. Such the stretchable platform was realized by transfer-printing of the as-prepared EMG electrodes on a $SiO_2$ wafer to a polydimethylsiloxane (PDMS) elastomer substrate. The epidermal EMG patch has skin-like properties owing to its unique mechanical characteristics: i) location on a neutral mechanical plane that enables high flexibility, ii) wavy design that allows for high stretchability. We demonstrated wireless EMG monitoring using our skin-attachable and stretchable EMG patch sensor integrated with the miniaturized wireless system modules.

**Keywords:** Bio-medical engineering; Gait disturbance; walking aids; Exoskeleton suit; Stretchable EMG sensors; Skin-attachable EMG sensor; Wireless health monitoring

---

## 1. Introduction

With a constantly aging society, there is a rising demand for new technologies that can manage and monitor health conditions. Advances in biomedical healthcare technology for the elderly and the disabled are rapidly taking place in accordance with these needs. In particular, aging phenomena like muscle weakness and decreased motor skills, or physical disorders such as stroke-induced hemiparalysis are very common in modern-day adults, and are increasingly occurring as the average life span is extended. These diseases cause chronic discomfort in daily behavior and walking, and they pose a potential risk of causing unexpected situations such as falling while in motion or other accidents in daily life. As a result, there is a growing need to develop walking-assisting technologies for the

disabled [1–20]. To implement these platforms, the development of skin-attachable and stretchable sensors for detecting and monitoring muscle signals involved in ambulation is essential (Figure 1).

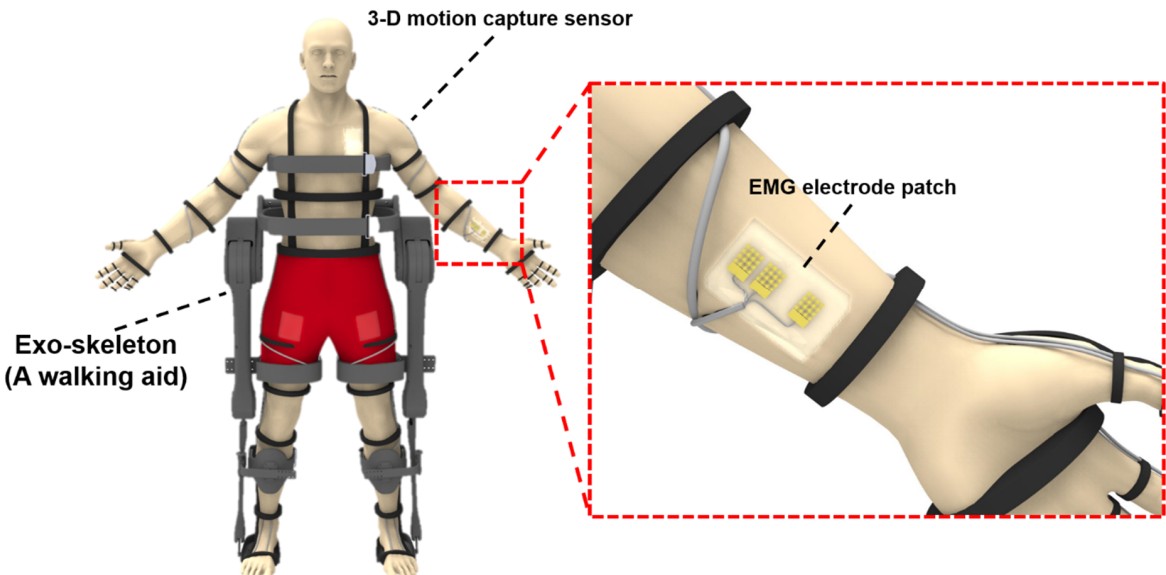

**Figure 1.** Schematic of a walking aid exoskeleton and the skin-attachable electromyographic (EMG) electrode patch.

Stretchable electronics that are implemented on soft and stretchable elastomeric substrates can be applicable to wearable sensor applications that can be worn or attached to the human body since the mechanical properties of the devices are similar to that of the human skin, allowing long-term use without any side effects such as feeling of irritation or inflammation when used [21–56]. Manufacturing technologies of these stretchable devices are almost compatible with conventional microfabrication for silicon-based electronics, and for this reason, it has been possible to implement various stretchable and wearable system on soft and stretchable elastomer substrate by using transfer printing method after being processed on a silicon substrate [57–59]. These stretchable devices can also operate safely with the elongation of the human skin during deformation [21–56]. To date, a lot of research has been proposed to develop stretchable electrode patches to detect the EMG, while it is attached to the skin [21–23,54], and also, there are several commercialized products of the EMG sensor, which have been developed. However, conventional EMG products do not have an intrinsically robust adhesion with the skin, so they easily slip off with the slightest movement when attached to the human skin. Furthermore, the noise they induce from the movement is severe, resulting in poor signal acquisition characteristics. For this reason, conventional EMG sensors require the application of foam tape or conductive gel to the adhesive surface of the devices to complement the adhesion property and signal acquisition performance of the skin to ensure the practical level of sensor capabilities [60–65]. One clear limitation of current technologies is the fact that the devices can only be used once, and not repeatedly. In addition, conventional measuring instruments for EMG are very large and heavy, and are wired to computers for operating EMG monitoring software, making it impossible for use in daily life.

Here, we described a stretchable EMG with signal acquisition performance near a commercial product level without the need to apply foam tape or conductive gel to surface. Metal-based stretchable electrodes were fabricated on $SiO_2$ wafer and transferred on soft and stretchable elastomeric film. Our devices are processed by conventional microfabrication applied to semiconductors and MEMS technologies so that metal electrodes with fine pattern can be elaborated into the desired shape and dimension, and multiple elements can be processed at high yield on wafer-scale substrate [21–35]. In addition, the interconnection between each electrode pattern consists of a serpentine structure, allowing the devices stretchability with mechanical deformation under strain [21–35]. The fabricated

metal electrode was transfer-printed on a soft and stretchable PDMS substrate with tens of micrometer thickness to implement stretchable and skin-attachable EMG sensors that can be applied to the human skin [21–23]. Our sensors adhere along the curvature of the skin and can be stretched so that they are not only robustly attached to the skin during movement, but also can reliably detect EMGs. We measured the mechanical and electrical characteristics of the developed stretchable electrode to evaluate the resistance under strain and impedance characteristics, signal acquisition performance, etc. Next, we applied the stretchable EMG sensor using our skin-attachable electrode patches to the human forearm, connected to a commercial EMG instrumentation device, and conducted EMG recording experiments. Finally, we have implemented a wireless EMG monitoring system connecting our EMG sensors with our own-designed wireless EMG module and conducted EMG monitoring experiments.

## 2. Materials and Methods

### 2.1. Fabrication and Transfer Printing of Stretchable EMG Electrode

Figure 2a shows the fabrication process of the stretchable EMG electrode. Each electrode component has a 2 mm × 2 mm square structure, and 4 × 4 array of components that constitutes an EMG device. The interconnection of each electrode consists of a serpentine structure, granting stretchability to the entire system by mechanical deformation of this structure under strain. The fabrication of metal-based stretchable electrodes was performed on $SiO_2$ wafer substrates, the metal electrodes are bi-layer structures consisting of titanium (Ti) adhesion layer and gold (Au) electrode layer. The top and bottom sides of the electrodes are supported with polyimide (PI) backbone layer, respectively, to separate the electrode from wafer substrates. PI (Poly(pyromelitic dihydride-co-4,4'-oxydianiline), amic acid solution 12.8 wt. %, SIGMA-ALDRICH) solution was coated uniformly on the treated substrate using a spin coater (EF-80P, E-FLEX, Co., Ltd.) and baked at oven. After the PI backbone bottom layer was formed on the wafer, negative photoresist (PR) (DNR L300-30, DONJIN SEMICHEM, Co., Ltd.) was spin-coated and photolithography process was carried out at mask aligner (MA6, SUSS MicroTec, Co., Ltd.) to draw electrode patterns on the substrate. Developer (AZ MIF-300, OCI, Co. Ltd.) was used to remove a non-exposed PR, then the electrode patterns were formed. Next, the Ti/Au metal layers were deposited by e-beam evaporator (EI-5K, ULVAC Technologies, Inc.) and the wafer was immersed into the acetone (OCI Co., Ltd.) to lift off all undeveloped PRs, leaving only Ti/Au metal layer of the electrode pattern on substrate. A PI layer was coated and baked once more to encapsulate the electrode pattern. Subsequently, the same PR used previously was coated and the photolithography process was performed under the same conditions as the previous step to draw patterns of the PI backbone layer and open the sensing area of the electrode layer. The patterned wafer was deposited by aluminum (Al) layer and immersed into acetone to lift-off, so that the Al layer was only remained on top of the PI backbone pattern and the PI area was exposed to the rest of the surface. Reactive ion etching (IPL2000E, DAEDING HIGH TECHNOLOGIES, Co., Ltd.) process was performed to remove the exposed PI area. During the process, the remaining Al layer on the PI backbone pattern and Ti/Au electrode layer functions as an etching mask to protect the PI layer below. After etching, the wafer was immersed into wet etchant (APAL-1, APCT, Co., Ltd.) to remove the remaining Al layer, then the microfabrication process of electrode devices was completed. Additional details about fabrication process were described in Supplementary note 1. After the fabrication of the devices was finished, the transfer-printing method was applied to move the devices from wafer to stretchable PDMS substrate. A piece of Teflon tape (903UL, Nitto Co., Ltd.) was attached to the slide glass so that the substrate can be separated into glass and PDMS (Sylgard 184, Dow silicones Co., Ltd.) solution in the ratio of 40:1 (precursor: curing agent) was drop-casted uniformly and cured. Using Water-soluble tape (No. 5414, 3M Co., Ltd.), the stretchable EMG electrode was separated from the wafer and moved onto the PDMS substrate. The water-soluble tape was dissolved by DI water, then transfer-printing of the stretchable EMG electrode to the elastomer substrate was finished. Figure 2b showed each component layer of

the finished stretchable EMG electrode patch, and Figure 2c presented a photographic image of the electrode sample.

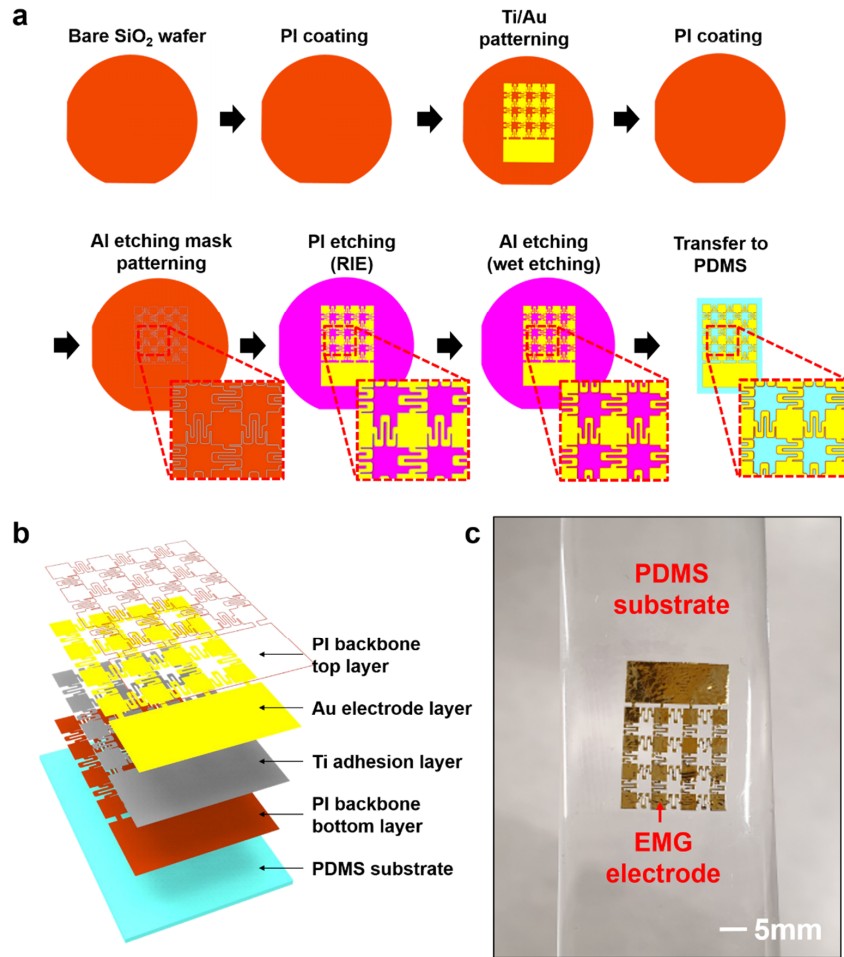

**Figure 2.** Schematic illustrations and photographic image of the ultrathin, stretchable electrode patch for EMG monitoring: (**a**) Fabrication process of the metal-based stretchable EMG electrode; (**b**) Each component layer of the device; (**c**) A photographic image of a stretchable EMG electrode patch.

### 2.2. Mechanical and Electrical Characterizations of the Stretchable EMG Electrodes

Stretching tests for measuring the mechanical and electrical performance of stretchable EMG electrode were performed using the following methods. The PET film was attached to the auto-stretching stage (Motorizer X-translation Stages, Jaeil optical system) to load the sample on the stage. Next, the stretchable EMG electrode sample was loaded on the stretching stage and fixed firmly using silicon epoxy (Sil-poxy, Smooth-On Co., Ltd.). To measure the electrical performance of the stretchable EMG electrode, a source measure unit (Keithley 2450 SourceMeter, TEKTRONIX, Inc.) was used. With a current of 0.5 mA applied to the sample, the resistance of the stretchable electrode was measured according to the elongation in real-time using the LabVIEW (National Instrument) program. Liquid metal (Gallium Indium eutectic, Sigma Aldrich) was used to connect the equipment with stretchable electrode.

### 2.3. In Vitro Demonstration of the Electrical Performance of the Stretchable EMG Electrodes

#### 2.3.1. Electrochemical Impedance Characterization

To characterize the electrochemical impedance property of the stretchable EMG electrode, electrochemical impedance spectroscopy (EIS) method was done by using a potentiostat (Versa

STAT3, AMETEK Inc.). A three-electrode system was utilized to measure the impedance of the electrode. A saturated calomel electrode and a platinum (Pt) wire (RDE0021, AT Frontier Co., Ltd.) was used as a reference electrode and a counter electrode respectively. The impedance of EMG electrodes in a phosphate-buffered saline (PBS) (1× PBS, SAMSHUN PURE CHEMICAL Co., Ltd.) solution was measured over the frequency range of 1Hz to 100kHz. The amplitude of applied voltage was 10mV root-mean-square (RMS) for potentiostatic EIS measurement.

### 2.3.2. Assessment of Signal Acquisition Performance

Before recording EMG signal by attaching the stretchable patch sensor to human skin, we conducted a test to verify the signal acquisition performance of the electrode in the in vitro environment. The measurement system was set for evaluation of signal acquisition performance. A reference electrode made by depositing Ti/Au on PET sheet, a commercial EMG electrode (EKG Ag/AgCl electrode 2223H, 3M Co., Ltd.) and a stretchable EMG electrode were immersed into the PBS solution. The two output terminals of a function generator (AFG3022C, TEKTRONIX, Inc.) were each connected to a wire immersed into PBS solution, and a oscilloscope (DPO4104, TEKTRONIX) was connected with both stretchable EMG electrode and commercial EMG electrode. The function generator applied input voltage on sine shape wave to both electrodes and the oscilloscope monitored the output signal acquired for each electrode.

### 2.4. In vivo EMG Recording

### 2.4.1. Sample Preparation for Epidermal EMG Sensor

In order to measure the EMG signals of the human skin, we manufactured epidermal electrode patches following the proposed method. PDMS solution with 10:1 ratio was mixed for more than five minutes. Evenly mixed PDMS solution was drop-casted on slide glass, spin-coated at 1000 rpm for 40 s and heated at 100 °C for 1 h. A water-soluble PVA film for delaminating epidermal patch sensor was covered on PDMS layer. Another PDMS solution mixed with 40:1 ratio was drop-casted on PVA film, spin-coated at 1000 rpm for 40 s and heated at 100 °C for 1 h. The stretchable EMG electrode fabricated on $SiO_2$ wafer was separated by a water-soluble tape and transferred on the PDMS substrate with 40:1 ratio. The water-soluble tape was dissolved by DI water, then preparation of skin-attachable and stretchable epidermal EMG patch sensor was completed.

### 2.4.2. Assessment of the EMG Recording Performance

The EMG monitoring experiment was performed by interconnecting the device with a commercial EMG measuring instrument (MP150, Bio-pac systems, Inc.). For the interconnection of the EMG patch sensor with measuring modules, the back end of the stretchable EMG patch was connected with one side of a conductive film where the other side was connected with PET sheet deposited by Ti/Au metal layer by heat. Metal-deposited PET sheet was connected to a wireless transmitter (BN-TX RSPEC-4.3, Bio-pac systems, Inc.) of the commercial EMG measuring instrument. Three EMG patch sample were connected to measuring module in the same way. Prepared epidermal EMG sensors were attached to human forearm and the EMG signal of several movements was measured. The measured signal from the sensor was transmitted from the wireless transmitter to the data acquisition unit (MP-150, Bio-pac systems, Inc.) connected with a PC, then recorded data was displayed on the EMG monitoring program.

### 2.5. Wireless EMG Monitoring

The wireless EMG recording experiment was conducted by interlocking epidermal EMG patch sensor with the self-development wireless EMG measuring module. Among the epidermal EMG patch sensor based on three-electrode method, two electrodes were attached to soleus and one electrode was attached to malleolus of less muscle. The three attached EMG sensors were connected to the measuring

module through the wire respectively. Wireless module was fixed with bandage to minimize shaking and impact by leg movement, and the interconnection between the epidermal EMG patch sensor and the wire was also secured with a Tegaderm film (1626W, 3M, Co., Ltd.) to minimize shaking of wire. The measured EMG signal of leg movement at soleus was transmitted to the receiver module connected a PC through wireless EMG module, then the recorded data was monitored on the EMG monitoring program.

## 3. Results

### 3.1. Mechanical and Electrical Properties of the Stretchable EMG Electrodes

The mechanical and electrical performance of stretchable EMG electrode is shown in Figure 3 and Figure S1. In Figure 3a, the stretchable EMG electrode shows stable mechanical characteristics as the serpentine structure between the square patterns fold out during the stretched pristine to 30 % stretching, which is the maximum tensile deformation range of human skin [66,67]. The stretchable EMG electrode not only had stable mechanical properties up to 50 % stretching, but also the resistance of the stretchable EMG electrodes was lower at 4 Ω due to the serpentine structure. Furthermore, the stretchable EMG electrode had a little increase in resistance compared to the pristine one, as seen in Figure 3b. Subsequently, we verified whether the stretchable EMG patch maintains stable mechanical and electrical performance in the cyclic stretching test at 30 %. The frequency of the stretching cycle was 6 cycles per minute. The result showed that the stretchable EMG electrode maintained stable mechanical performance under a 100 cyclic stretching test (Figure 3c). During the cyclic test, the ratio of stretched electrode resistance against initial resistance remained almost 1, which verified that stretchable patch had very stable electrical property. As a result, the stretchable EMG electrode patches was verified to have excellent mechanical properties and stable electrical characteristics under strain.

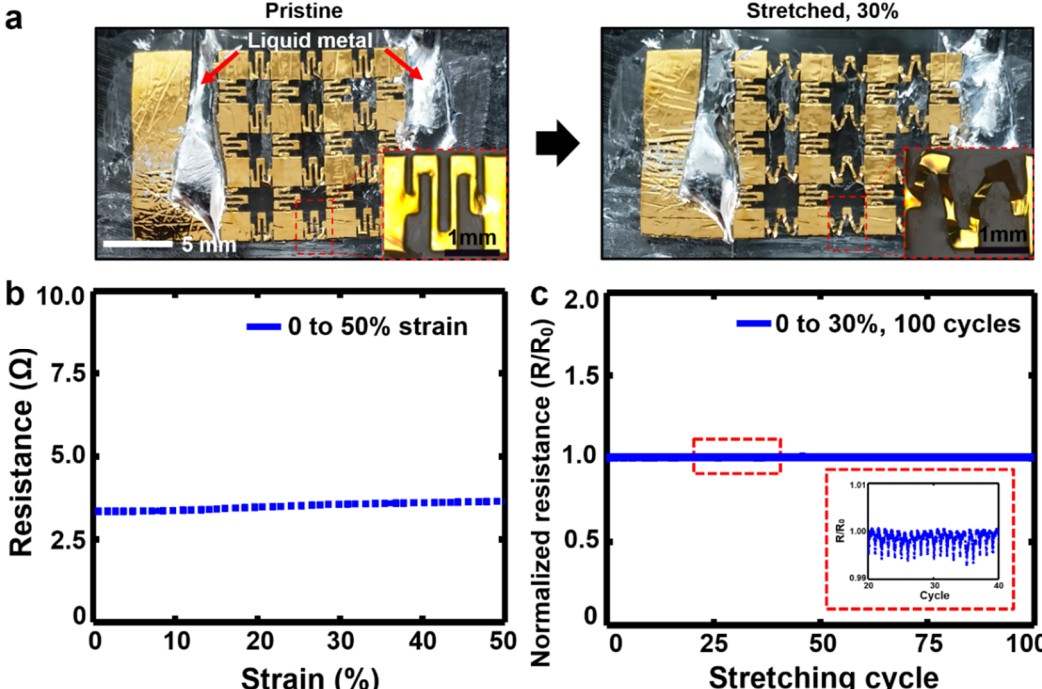

**Figure 3.** Mechanical stretchability and electrical stability of the stretchable EMG electrode: (**a**) Images of a pristine electrode (left) and a stretched electrode (right). Corresponding microscopic images of stretched serpentine structure (Inset); (**b**) Resistance-strain characteristics of the stretchable EMG electrode; (**c**) Normalized resistance during 100 stretching cycles. A graph of partially enlarged section (Inset).

### 3.2. In Vitro Electrical Performance of the Stretchable EMG Electrodes

Quantifying the impedance and signal acquisition property is necessary to assess the electrochemical and electrical performances of the electrode before *in vivo* tests. Therefore, several tests were conducted to evaluate the performance of the stretchable EMG electrode in the *in vitro* experiment prior to applying the devices to human body (Figure 4). The electrochemical impedance characteristics of the stretchable EMG electrode patch immersed in PBS solution were measured (Figure 4a). The impedance of the electrode sample with a total area of 96.29 mm² at 1 kHz was 1.8 kΩ (Figure 4b). This value indicates that our electrodes had sufficiently low impedance characteristics needed to obtain vital signs. To assess the signal acquisition performance in the PBS solution, various amplitude of sine waves with a frequency of 1 kHz set as a function generator was applied to a stretchable electrode sample and a commercial EMG electrode respectively, and the acquired signal on each of the electrodes was displayed on an oscilloscope and compared to each other. (Figure 4c,d). As a result of varying the peak-to-peak size of the input signal to 1 V, 3 V, 5 V, 7 V, and 10 V, the magnitude of the acquired voltage at the stretchable EMG electrode patch had a peak-to-peak signal size of at least 90 % to about 100 % of those of the commercial electrode (Figure 4e). Based on these results, the signal acquisition performance of the stretchable electrode patch was verified to be close to that of commercial electrodes.

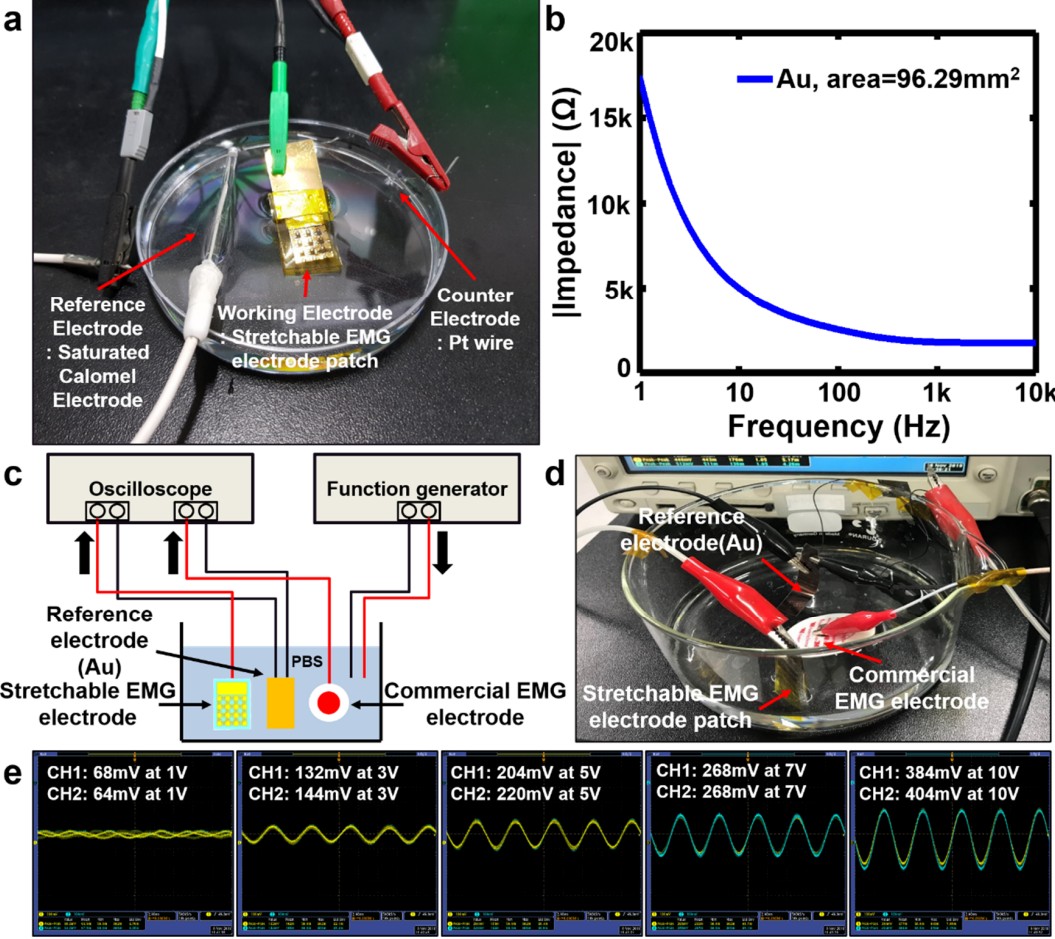

**Figure 4.** In vitro electrical characterization of the stretchable EMG electrode: (**a**) Experimental setup for impedance measurement; (**b**) Electrochemical impedance property of the device according to frequency; (**c**) The schematic illustration of experimental setup for evaluation of signal acquisition performance; (**d**) Sample configuration for comparison of signal acquisition performance; (**e**) Results of acquired output signal for a developed stretchable electrodes (CH1) and commercial electrodes (CH2).

### 3.3. In Vivo Vital Sign Recording Using the Stretchable EMG Sensor

After evaluating the *in vitro* electrical characteristics of the stretchable EMG electrode patch in PBS solution, the epidermal EMG sensor was applied to the human body to perform EMG recording experiment. Figure 5 shows the experimental results of EMG recording with epidermal EMG sensors which were ultra-thin enough to be attached to the skin. The EMG patches were attached to the subject's forearm and EMG signals for several movements were measured. The EMG recording performance of the epidermal EMG sensor was compared with commercial EMG sensors with foam tape and sticky gel (Figure S2). The developed EMG sensor and commercial sensor were configured by 3-electrode system to operate one electrode as a reference electrode and two electrodes as sensing electrodes, respectively (Figure 5a,e). EMG signals for the standby state (Figure 5b,f), fist-clenching (Figure 5c,g) and wrist bending (Figure 5d,h) situation were measured and each movement was performed three times in succession. The signal acquisition performance for each sensor was evaluated by calculating the average signal-to-noise ratio (SNR) of the voltage amplitude for the signal interval in motion situation. SNR values of each situation were calculated using the expressions in Equation (1).

$$ \text{SNR} = 20\log\frac{V_{signal}}{V_{noise}}\ (dB) \tag{1} $$

For the commercial EMG sensors, the average SNR of EMGs in clenching a fist and bending a wrist motion was calculated to be 12.1 dB and 10.2 dB respectively (Figure 5c,d). The average SNR of EMGs for the epidermal sensor in the same situation was 10.7 dB and 9.89 dB, respectively (Figure 5g,h). These results indicated that SNRs of the epidermal EMG sensor were about 88% in fist grip movement and 97% in wrist flexion movement compared to those of commercial gel-type sensor. The surface of PDMS substrate for the proposed devices was very sticky due to its composition ratio of 40:1 for the precursor to curing agent, and that spin-coated layer had an ultra-thin thickness of less than 100 micrometers. Thus, our epidermal EMG sensor was able to adhere to the skin without applying foam tape or sticky gel, and the signal acquisition performance of proposed skin-attachable and stretchable EMG sensor was close to the level of commercial product.

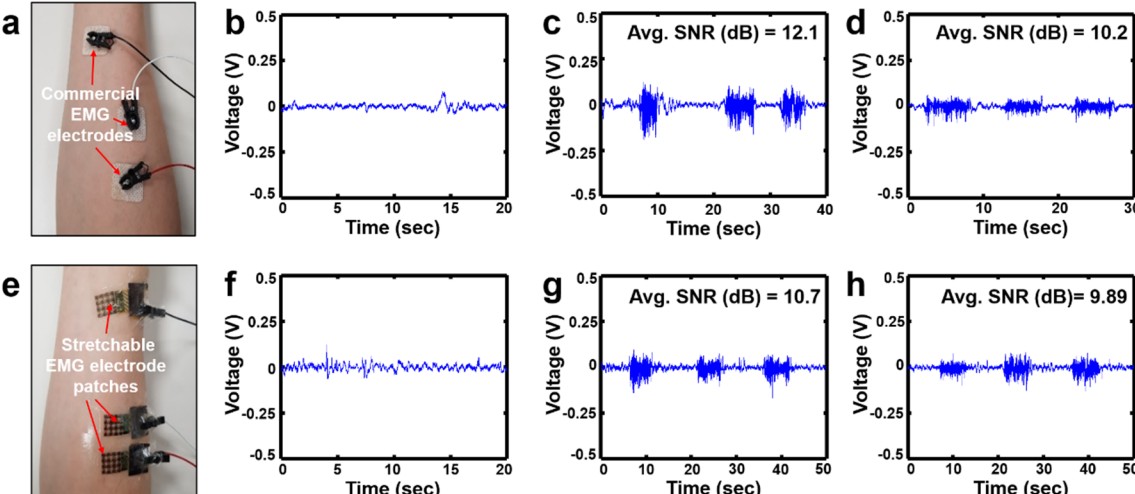

**Figure 5.** Results of in vivo EMG recording with the epidermal sensors and conventional sensors: (**a**) Commercial EMG sensors attached to a forearm; (**b**) Background signal of commercial sensors in standby state; (**c**) Activated EMG signals acquired by commercial sensors in clenching a fist motions; (**d**) Activated EMG signals acquired by commercial sensors in bending a wrist motions; (**e**) Epidermal EMG sensors attached to a forearm; (**f**) Background signal of epidermal sensors in standby state; (**g**) Activated EMG signals acquired by epidermal sensors in clenching a fist motions; (**h**) Activated EMG signals acquired by epidermal sensors in bending a wrist motions.

### 3.4. Demonstration of Wireless EMG Signal Monitoring

Applying the developed skin-attachable and stretchable EMG patch sensor to human body, we demonstrated that the signal acquisition performance for vital sign corresponds to that of the commercial EMG sensor. Finally, we conducted a wireless EMG recording experiment by connecting our EMG sensor with the newly designed wireless EMG module (Figure 6). Figure 6a shows the schematic diagram of the self-designed wireless EMG module and the process of wireless EMG monitoring. The developed wireless EMG module consists of Analog-to-Digital Converter (ADC, ADS1299-4), data processor (MCU, STM32 F405RG), and Bluetooth (FB155BC). It can detect up to four channels of EMG signals simultaneously, and acquired EMG signals are converted to digital data by ADC at a sampling rate of 1kHz for each channel. After signal processing and allocating data packets from the MCU, processed data is wirelessly transmitted via Bluetooth to the receiver module connected to the PC. The module is equipped with a battery with a capacity of 500 mAh, enabling up to 5 h of wireless operation. The height, width, thickness, and weight of the module are 40 mm, 40 mm, 6.2 mm, and 7.6 g, respectively, which functions as a portable EMG measuring instrument that is very small and light-weight (Table S1). EMG data acquired via the receiver module to the PC was monitored through the GUI of the designed software and recorded in real-time. During monitoring, Fast Fourier Transform (FFT) signals were simultaneously displayed so that it could be immediately determined whether the signals displayed were EMG signals or noise signals. Figure 6b shows a photo-graphical image of the experimental set-up of wireless EMG monitoring and wireless EMG module. To activate the EMG of the soleus where the sensor was attached, calf raise exercise was performed five consecutive times at intervals of about 2 s and the corresponding EMG signals were monitored (Figure 6c). The epidermal EMG sensor attached to the soleus was well adhered to the skin that no slip between the skin and the electrode occurred during the calf raise movement, resulting in no baseline fluctuation noise. Background's signal was shown to be about $3.0 \times 10$ mV or less, and the average SNR of the acquired EMG signal was calculated to be 15.77 dB according to the Equation (1) (Figure 6d).

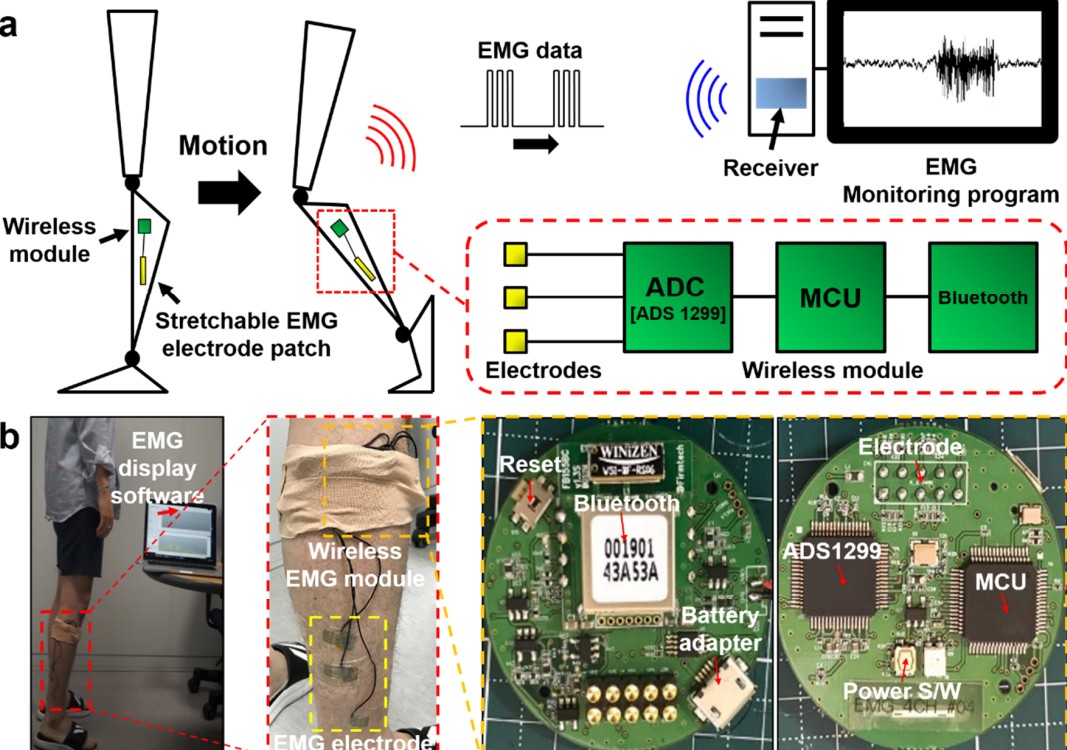

**Figure 6.** *Cont.*

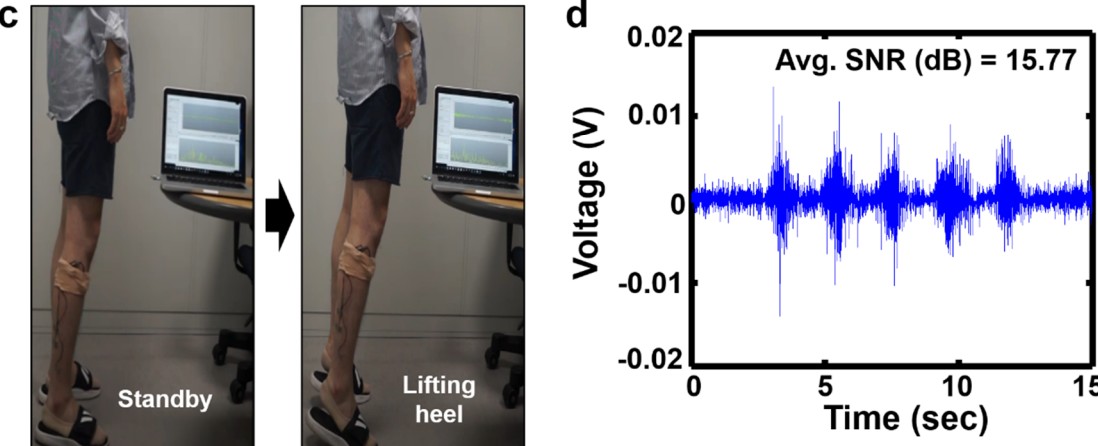

**Figure 6.** Results of wireless EMG by connecting the epidermal sensors with the wireless module: (**a**) The Schematic illustration of wireless EMG module and monitoring process; (**b**) The Experimental setup of wireless EMG monitoring and photographic images of the wireless EMG module; (**c**) Calf lift motions to activate EMG of soleus; (**d**) Activated EMG signals of soleus acquired by wireless epidermal sensing system in calf lift motions.

## 4. Discussion and Conclusion

In this study, we implemented the metal-based stretchable EMG electrodes by the microfabrication process and transfer-printing method. The devices showed stable resistance even with repetitive elongation, similar the maximum tensile range as found in human skin. Furthermore, the devices had low impedance and signal acquisition performance similar to that of commercial electrodes. The skin attachable and stretchable EMG sensors which had a great level of SNR performance for EMG signal close to that of commercial EMG electrodes were transferred onto an ultra-thin elastomer substrate made of spin coating and thus, was tightly attached to the skin without the application of foam tape or conductive gel. With the proposed design, the wireless EMG module was made to be light weighted and small in size to aid practicality and ease of use for the daily activity. As a result, the wireless EMG monitoring was demonstrated effectively by connecting the developed EMG sensor and wireless module. In further works to realize practical walking aid platforms, signal acquisition performance of the skin attachable sensor for wireless vital sign monitoring must be improved. Adhesion to the skin also should be more robust to minimize noise generation by movement. Furthermore, practicality and portability of wireless EMG monitoring system should be improved by the realization of low power, miniaturization and the reduction of weight of the wireless module, for effective processing and communication of the bio-data detected by the wearable sensors. Ultimately, walking aid technologies should be implemented as soft exosuit devices, more advanced from the conventional rigid exoskeleton to maximize user convenience. The results of this study therefore show the potential for the development of skin-attachable sensors and wireless health monitoring techniques applicable to walking aids that can help the elderly and disabled with weakened muscles in their daily lives.

**Supplementary Materials:** The following are available online at http://www.mdpi.com/2079-9292/9/2/269/s1, Figure S1: Resistance characteristics of the stretchale electrode patch under strain condition occurring delamination. Figure S2: The experimental setup of in-vivo electromyogram (EMG) recording: (a) commercial gel-type EMG sensors and a measuring instrumentation. title, Table S1: The specification of the wireless EMG module.

**Author Contributions:** Conceptualization, D.S.; Data curation, S.L. (Sangkyu Lee), J.Y. and D.L.; Formal analysis, S.L. (Sangkyu Lee), J.Y., D.L., D.S. and S.L. (Sangkyu Lee); Funding acquisition, J.C., J.K. and D.S.; Investigation, S.L. (Sangkyu Lee), J.Y., D.L., D.S. and S.L. (Sangkyu Lee); Methodology, S.L. (Sangkyu Lee) and D.L.; Project administration, J.C., J.K., S.L. (Sangyoup Lee) and D.S.; Resources, S.L. (Sangkyu Lee), J.K., S.L. (Sangyoup Lee) and D.S.; Software, D.L.; Supervision, K.J.Y., S.L. (Sangyoup Lee) and D.S.; Validation, S.L. (Sangkyu Lee), J.Y. and D.L.; Visualization, S.L. (Sangkyu Lee), J.Y., D.L., D.S. and M.J.; Writing – original draft, S.L. (Sangkyu Lee), J.Y.

and D.L.; Writing – review & editing, S.L. (Sangkyu Lee), J.Y. and D.S. Sungjun Lee, Jiyong Yoon and Daewoong Lee equally contributed to this work. All authors have read and agreed to the published version of the manuscript.

**Funding:** This research was supported by the future source technology development program for bio-medical engineering funded by the Korea Institute of Science and Technology (KIST) (No. 2E29680).

**Conflicts of Interest:** The authors declare no conflict of interest.

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
