# Peer review of "Wireless Epidermal Electromyogram Sensing System"

_electronics, doi:10.3390/electronics9020269_

Round 1
Reviewer 1 Report
Author report on the fabrication of stretchable electrodes for the development of walking ads platform. Stretchable electronics is plenty of works dealing with original process. That’s why an important remark of the reviewer is to improve the introduction part dealing with other works who provide other printing transfer technologies. Other inconsistencies and additional works has to be added in the revised manuscript to be published in electronics.
Concerning the introduction part, authors do not report about processing allowing to transfer stretchable interconnects onto PDMS, textile… As the processing part is an important result of this work, authors have to include this point in the introduction part. They should cite the following works, because it reports on oher original process:
- Rogel, R., et al.. (2017). Spontaneous buckling of multiaxially flexible and stretchable interconnects using PDMS/fibrous composite substrates. Advanced Materials Interfaces, 4(3), 1600946.
- Linghu, C., et al. (2018). Transfer printing techniques for flexible and stretchable inorganic electronics. npj Flexible Electronics, 2(1), 26.
- Lu, N., et al. (2019). Water Transfer Printing Enhanced by Water‐Induced Pattern Expansion: Toward Large‐Area 3D Electronics. Advanced Materials Technologies, 4(4), 1800600.
Results will be more impressive if authors studied stretching in y and x directions at the same time, as the serpentine interconnects theoretically allow to reach multiaxial stretching.
Authors should indicate a figure highlighting the strain limit (Resistance value as function of strain since the strain limit where the resistance value dramatically increased)
In figure 3c what is the frequency of the stretching cycle?
In materials and methods section (2.1 “fabrication and transfer printing of EMG electrode”): Reviewer thinks that the process description is difficult to follow due to the high amount of experimental details. These details are essential to reproduce the process however, authors should summarized the main process steps in this section and moved the additional experimental details in a “supplementary information file”.
In section 3.2, the reviewer did not understand why in vivo test must be conducted in PBS solution. Author should develop this section.
Abstract length should be reduced (approximately 200 words).

Author Response
We appreciate the reviewer for pointing out the key advancements made in this work and for recommending our work for publishing in MDPI Electronics. All the questions and suggestions have been fully addressed in the point-to-point response below, and all the associated revisions are highlighted in the paper.
please check the attached file.
Thank you.
Yours sincerely, Mr. Lee.

Reviewer 2 Report
The paper describes an interesting and important development of a flexible EMG system. However, there are comments in the description that not fully are in agreement with the common state of the art within the are. For example, standard electrodes for ECG and EMG recordings have a foam tape that means that gel not is needed (also the electrode used for comparisons in the tests). Also, many of the references are very old, and thus the state of the art does not describe the current development properly. Also, the elderly and disabled are treated as one group, needing the same solutions.
The photographs in Figure 4 and 6 do not in a clear way present the experimental set up. It could be clearer with drawings in some of them.
The English languish needs to be improved. For example, there are many long sentences that are very difficult to understand and that are incomplete.
Author Response

(The authors gave the same response as above.)

Round 2
Reviewer 1 Report
This work is worthy to be published as it.
Reviewer 2 Report
The authors have considered most of my comments.